# Analyzing the Development Possibilities of the Mountain Area of Banat, Caras-Severin County

**Paula-Diana Peev-Otiman** [1] **and Nicoleta Mateoc-Sîrb** [1,2,*]

1 Department of Management and Rural Development, Banat's University of Agricultural Sciences and Veterinary Medicine "King Michael I of Romania" from Timisoara, 300645 Timisoara, Romania
2 Romanian Academy, Timisoara Branch, Research Center for Sustainable Rural Development of Romania, 300223 Timisoara, Romania
* Correspondence: nicoletamateocsirb@usab-tm.ro

**Abstract:** A county such as Caras-Severin in the mountain area of Banat, with extraordinary natural tourism potential, has a real chance for tourism development only under the conditions of a strong economy and an infrastructure that facilitates and supports tourism activity. In turn, tourism, as an economic activity generating jobs and new added value, must contribute, through feedback, to the general economic development of the county. This research offers a case study on the possibilities of tourism development of these areas by exploiting the available natural and anthropic potential. The economic and social development of such a county, including from a tourism point of view, is strictly linked to the general economic development and evolution of Romania, both in the national and international context. To position the contribution of tourism to the development of Caras-Severin County as correctly as possible, we briefly present the general economic situation, including tourism, of Romania, by comparing it with the neighboring and, at the same time, competing countries from a tourism point of view—Bulgaria, Croatia, Slovenia and Serbia,—as well as with counties in Romania similar to Caras-Severin County. In terms of its general economic development level, Caras-Severin County is among the least developed counties in the country. In this situation, the development of specific forms of tourism is justified to contribute to the economic development of the area.

**Keywords:** sustainable tourism; rural area; analysis; development possibilities

## 1. Introduction

Rural areas invest in tourism [1–4] to diversify their economies, which is necessary for growth, employment and sustainable development [5–7]. They offer real possibilities [8–12], either as attractive places to live, or as reservoirs of natural resources and landscapes of great value [13]. In this context, it is necessary to ensure the coherence of community policies, create synergies between them, preserve the natural environment and protect rural areas. Mountain areas require prospective rural development policies which integrate both the conservation requirements of this special natural environment, as well as the sustainable well-being of the inhabitants and unique products [14].

The sustainable development of local communities has as its major objectives protecting the environment, fighting poverty, improving the quality of life, and developing and maintaining a viable and efficient local economy. The concept of sustainable development supposes [15–17] performance on three levels: economic—increasing the degree of the capitalization of resources; ecological—recycling, avoiding environmental degradation and reducing the removal of fertile land from the agricultural circuit; and social—increasing the number of jobs, practicing some traditional jobs and attracting the population to practice tourism.

Sustainable tourism includes the development of all forms of tourism [18–20] through the capitalization of natural and cultural resources (without damaging the environment),

and through tourism management and marketing that respects the natural, social and economic integrity of the environment [21,22] in the form of innovative sustainable development [23–25]. The emergence in practice of some forms of sustainable tourism has resulted from the need to protect the natural, social and cultural wealth that constitutes the common heritage of humanity, but also to satisfy the needs of tourists and the local population. The sustainable development of tourism is realized through forms of tourism which are based on the following principles [26–30]: reducing the impact of tourist activity on the natural environment in order to achieve ecological sustainability, and contributing to the maintenance and improvement of the conservation state by returning a part of the income to actions regarding environment protection; reducing the negative impact of tourist activity on the local community and its members in order to achieve social sustainability by developing those forms of tourism that do not disrupt and do not interrupt the daily lives of the population in tourist destinations; minimizing the negative impact of tourism on the culture, traditions and customs of local communities in order to achieve cultural sustainability through the development of tourism capable of maintaining the authenticity and individuality of local cultures; maximizing the economic benefits at the level of the local community as a result of the tourism development in order to obtain economic sustainability, which constitutes one of the most important principles of sustainable development, putting tourism at the service of the protection and economic development of local communities and protected areas; educating, training and informing the tourist to improve personal attitudes towards the environment and reduce its negative impact, which includes an ecological–educational component for visitors, locals, the local administration and the local population; encouraging local control, a basic principle of sustainable development, according to which the local community participates in and is consulted in all aspects of sustainable tourism development, being an active decision-making factor, the key elements of which are the local ownership of tourism infrastructure elements (for example, accommodation structures) and the involvement of the community and local administrations.

The ability of tourist destinations to remain resilient against all emerging problems [31–33], to attract visitors for the first time and subsequently retain them, to remain culturally unique [34–36] and to be in permanent balance [37] with the surrounding environment, all fall within the context of sustainable tourism development [38–40]. The main objectives of sustainable tourism are the following: avoiding negative impacts on the environment, natural resources, tradition and local culture; ensuring the conservation of local ecosystems; generating income; creating new jobs; and increasing the country's visibility and competitiveness worldwide. Special forms of tourism, such as sports tourism, religious tourism, children's camps and educational visits, could further support each country's sustainable tourism efforts [41].

The World Economic Forum's Annual Meeting in Davos-Klosters, Switzerland, from 22–26 May 2022 highlighted two major risks for Romania's future: man-made environmental damage and crises related to employment and livelihood availability. In this context, we consider that agritourism, if executed correctly, is a type of tourism with a very low impact on the environment, as it allows for sustainable use and protection of resources. It is usually aimed at small groups of tourists, has educational purposes, all products consumed by tourists are sourced from the farms, it generates little waste and it allows the transition to a circular economy on the agritourism farm. In this context, we believe that agritourism should be the most promoted form of tourism in the future, as it has the lowest environmental impact by educating tourists about the protection of biodiversity and ecosystems in the areas they visit and thus contributing to climate change mitigation.

Sustainable tourism organizations strive to include local people in the tourism value chain and in the conservation of natural resources [42,43], which are ways to lift these communities out of poverty [44]. The tourism and hospitality sectors have the potential to employ residents, which indirectly contributes to economic growth and development, and provides income through job creation, including tour guides, hotel staff, etc. This

is particularly valuable when sustainable tourism is a promoter of entrepreneurship and small businesses, as well as of the empowerment of disadvantaged population groups, for example, youth and women (which is also directly linked to sustainable development goals (SDGs)).

The purpose of this paper is to identify the different forms of tourism that can be practiced in the mountain area of Banat, in Caras-Severin County, depending on the natural and anthropic resources available in the area, which will contribute to the development of the local business environment and which, through their consolidation in the long term, will generate positive economic–social externalities. In addition to generating economic growth and stimulating the development of local communities, sustainable rural tourism also represents travel opportunities for tourists from lower income groups, as this type of tourism does not charge very high taxes. At the same time, it offers tourists an alternative holiday while increasing the awareness and appreciation of traditional rural areas and protecting biodiversity and natural ecosystems.

In this sense, we have proposed the following objectives:

**OB 1:** *Economic and social analysis of Caras-Severin County, viewed in the context of the V West region, in a national context as well as that of the general economic state of Romania, and in a South-Eastern European regional context;*

**OB 2:** *Researching the current level of tourism in Caras-Severin County and in Romania and the dynamics of its development to increase its contribution to the GDP of Romania and Caras-Severin County.*

We consider the analysis and results of the present research to be timely for decision makers in the field of tourism because, so far in Romania, we are facing problems related to a lack of data, the limited capacity of decision makers and public policy makers to analyze and use data in decision making and, last but not least, certain shortcomings related to the collaboration between academia and the public administration when it comes to data and data processing.

Tourism as a factor in economic growth has not been taken into account by the Romanian government or by the directorates specifically designated by it. Perhaps the best example of this is the last exhibition by the Romanian Tourism Authority at the International Convention in New York, where they exhibited A4 images printed on a black canvas wall to promote tourism in our country. The lack of interest, demonstrated by the fact that no special governmental directorate assigned to tourism development has been created, and that there is no authority specialized for this purpose, is the biggest problem interfering with the development of tourism in this period. Because of this neglect, a domino effect is created, which ultimately leads to individuals investing in their own tourism businesses as they see fit without there being a national plan to coordinate tourism development in areas where there is tourism potential. This also leads to disruption of the landscape, fluctuating prices, often average or low standards, and other problems that are currently observed in the Romanian tourism industry.

## 2. Research Methodology

*2.1. Delimitation of the Researched Area—The Mountain Area of Banat, Caras-Severin County, Romania*

To delimit and rank tourist areas, it is necessary, first of all, to perform an inventory and to know all the components of their tourism potential, grouping them in space and then using quantitative and qualitative evaluation to establish their development opportunities, the development forms they can generate and the necessary equipment for efficient and competitive management.

To achieve the proposed objectives of the present paper, initially extensive bibliographic studies and reviews were carried out. From the studied bibliography, data were selected and considered related to the researched topic to deepen the proposed analysis. We resorted to an objective analysis of the problems, opportunities and tourism context of the area delimited for research. In this regard, we analyzed the economic–social situation

of Caras-Severin County, in a national and a regional context, and identified the natural, historical and cultural heritage that is the basis for the development of sustainable tourism in that area, considered as an area with a large potential in terms of developing local tourism and creating new jobs for the population.

Caras-Severin County has an area of 851,976 km$^2$, being, in terms of area, among the largest counties in the country and occupying the third position after Timis County (869,665 km$^2$) and Suceava (855,350 km$^2$). From a geographical point of view, Caras-Severin County is the "most mountainous" county in the country, with over 80% of the surface area being included in the mountain area, according to the regulations for the inclusion of Territorial Administrative Units (UAT) in the mountain area. The mountain structure of the county, conferred by the mountain massifs, and their mineral wealth have favored, over time, a complex economic development of mining, metallurgical industry, fruit growing, agropastoral activities, tourism, etc.

The massive structural changes in the county's economy (predominantly industrial, and mostly mono-industrial before 1990) caused important reductions in the employed population, with Caras-Severin County having one of the lowest percentages of employment. At the same time, Caras-Severin County "offers for export" some of the largest numbers of young and middle-aged women engaged in the home care of people in Austria, Italy, Germany, etc. More than 6000 women from the county work (monthly) in nursing abroad, and the effect of this "export" phenomenon is what gives Caras-Severin County the second highest rate of school dropouts in the country (3.4%).

The active population of the county, according to statistical data provided by the INS, has the following structure: agriculture, forestry and fishing 28%; industry 24%; administration, health and education 12.5%; trade 12.0%; services 8%; construction 7%; transport–storage 6%; and tourism, hotels and restaurants 2.5%. The extremely high share of the population employed in agriculture (for a non-agricultural county such as Caras-Severin County), being almost a third of the county's active population, is, from our point of view, questionable, considering the main characteristics of the county's agriculture being mainly subsistence and semi-subsistence.

The case of Caras-Severin and Banat is a classic case of a high potential–low development area, due to a series of factors, such as emigration, lack of infrastructure, lack of investments, etc. However, this specific area is indeed one of the utmost beauty and, some might say, one forgotten by the changes of time. Due to its geographical location, technology and development came here quite late, with the rural patriarchal life still being present in the first half of the last century. This has made it possible for many traditions and values to persist until today, with these representing a huge attraction for tourists, especially with agritourism and seeking the pastoral lifestyle being one of the biggest trends at the moment. This area could teach people to go back to some core human values, while generating profit from this.

In Caras-Severin County, according to the latest statistical data, the number of employees is 54,400, meaning that they are 19.8% of the total population of the county and 70.0% of the active population, which totals 77,700 employees and farmers combined (INS, 2019). A simple calculation shows that the active population (employees and farmers) represents only 28.3% of the total population of the county (275.2 thousand inhabitants), although the potential active population of the county is 150.2 thousand inhabitants (of which only 54.5% are in the current active population, counting those between school age and retirement age). From these data, there is an alarming demographic–occupational conclusion for Caras-Severin County, because it has one of the lowest employment indicators in terms of economic and social activities.

Another worrying demographic phenomenon for the general economy of the county is the depopulation of both urban and, especially, rural areas. During the period from 2007, the moment of accession to the EU, until now, the population of the county has decreased, through migration (particularly external) and naturally, by 70,700 inhabitants (346.9 thousand in 2007; 275.2 thousand in 2021).

The causes of this negative demographic phenomenon are many, but mainly it has been the particularly low general level of economic development, due to the absence of a strategy in line with the county's potential, and determined by a precarious economic and social conversion after 1990 and, in particular, the existence of successive county administrations (in almost all mandates after 1990) of poor quality, from all points of view: conception, involvement, concern, cooperation and collaboration with central and local authorities, etc.

The place of Caras-Severin County in Romania's economy and its level of economic development—with special reference to the development of tourism—are presented through the following synthetic economic indicators: GDP of the county; GDP/capita; and foreign investments and exports of the county for the period 2017–2019, in a national and a regional context (V West Development Region) (Table 1).

**Table 1.** The place of Caras-Severin County in Romania's GDP.

| Place | County | EUR (Millions) | | |
| --- | --- | --- | --- | --- |
| | | 2017 | 2018 | 2019 |
| 1 | Cluj | 9425 | 10,308 | 11,641 |
| 2 | Timis | 8525 | 9615 | 10,822 |
| 30 | Caras-Severin | 2087 | 2164 | 2408 |
| 41 | Covasna | 1404 | 1578 | 1742 |
| 42 | Giurgiu | 1195 | 1676 | 1499 |
| | Romania | 4319 | 4364 | 4841 |

Source: INS (consulted in November 2022) [45].

From an economic and social point of view, in Caras-Severin County a vicious economic cycle has manifest itself, unfortunately with maximum negative effect: poorly developed economy → economic underemployment of the workforce → even less developed economy → etc.

In terms of its general economic development level, Caras-Severin County is among the least developed counties in the country.

Regarding the gross domestic product per inhabitant, the main synthetic economic indicator that expresses the average yield or productivity of the employed population, Caras-Severin County is in 21st place, among the top counties, with EUR 8879, compared to EUR 10,666 which is the national average, or EUR 15,344, which is the value for Timis County (INS-2019 data). (Table 2).

**Table 2.** The place of Caras-Severin County in terms of GDP per inhabitant.

| Place | County | EUR | | |
| --- | --- | --- | --- | --- |
| | | 2017 | 2018 | 2019 |
| 1 | Cluj | 13,411 | 14,628 | 16,466 |
| 2 | Timis | 12,209 | 13,707 | 15,344 |
| 21 | Caras-Severin | 7478 | 7866 | 8873 |
| 41 | Covasna | 4823 | 8494 | 6182 |
| 42 | Giurgiu | 449 | 5026 | 5433 |
| | Romania | 7791 | 9560 | 10,666 |

Source: INS (consulted in November 2022) [45].

The weak economic position of Caras-Severin County is explained by the extremely poor economic and social conversion after 1990, its unattractiveness for investors, both

Romanian and foreign, and the low contribution of the county's economy to Romanian exports. (Table 3).

**Table 3.** The position of Caras-Severin County in foreign direct investment (million EUR).

| Place | County | 2017 | 2018 | 2019 |
|-------|--------|------|------|------|
| 1 | Ilfov | 4165 | 4451 | 5188 |
| 2 | Timis | 3998 | 4359 | 4386 |
| 30 | Caras-Severin | 287 | 162 | 196 |
| 41 | Mehedinti | 5 | 16 | 19 |
| 42 | Gorj | 4 | 3 | 3 |
| | Romania | 1709 | 1797 | 1965 |

Source: INS (consulted in November 2022) [45].

A county such as Caras-Severin, with such a large amount of natural resources (wealth) and exceptional tourist offerings, attracts "negligible" amounts of foreign investments, about EUR 200 million/year, compared to EUR 4386 million in the neighboring county, Timis, and the EUR 1965 million of the national average (ten times less, compared to the average county in the country, in 2019). (Table 4).

**Table 4.** The position of Caras-Severin County in exports (million EUR).

| Place | County | 2017 | 2018 | 2019 |
|-------|--------|------|------|------|
| 1 | Timis | 6292 | 6901 | 7307 |
| 2 | Arges | 6022 | 6240 | 6226 |
| 30 | Caras-Severin | 317 | 347 | 360 |
| 41 | Giurgiu | 76 | 85 | 73 |
| 42 | Gorj | 54 | 65 | 68 |
| | Romania | 1492 | 1612 | 1643 |

Source: INS (consulted in November 2022) [45].

Regarding the annual export of goods, Caras-Severin County is also in the 30th position among the country's counties. Caras-Severin County exports products worth EUR 360 million in a year, compared to EUR 6226 million for Timis County or the national average of EUR 1643 million (data from 2019). (Table 5).

**Table 5.** The position of Caras-Severin County by territorial area.

| Place | County | Km$^2$ |
|-------|--------|--------|
| 1 | Timis | 869,665 |
| 2 | Suceava | 855,350 |
| 3 | Caras-Severin | 851,976 |
| 41 | Giurgiu | 352,602 |
| 42 | Ilfov | 158,328 |

Source: INS (consulted in November 2022) [45].

In a regional context (V West Region), Caras-Severin County ranks at the lowest level for all economic indicators (Tables 6–9). The economic and social state of Caras-Severin County, viewed in the context of the V West region and in a national context, as well as the general economic state of Romania, analyzed in a South-Eastern European regional context, requires a deeper analysis of the causes of this situation of precarity for Romanian tourism, in general, and for Caras-Severin County in particular.

**Table 6.** The position of Caras-Severin County by population.

| Place | County | No. People |
|---|---|---|
| 1 | Iasi | 792,131 |
| 2 | Prahova | 725,515 |
| 3 | Cluj | 704,784 |
| 4 | Timis | 701,690 |
| 36 | Caras-Severin | 275,181 |
| 41 | Covasna | 203,504 |
| 42 | Tulcea | 195,626 |

Source: INS (consulted in November 2022) [45].

**Table 7.** Gross domestic product/county in Region V West (million EUR).

| Place | County | 2017 | 2018 | 2019 |
|---|---|---|---|---|
| 13 | Caraș-Severin | 2087 | 2164 | 2408 |
| 21 | Hunedoara | 2964 | 3236 | 8671 |
| 30 | Arad | 4128 | 4539 | 5055 |
| 41 | Timis | 8525 | 9615 | 10,822 |

Source: INS (consulted in November 2022) [45].

**Table 8.** Gross domestic product/inhabitant in Region V West (EUR).

| Place | County | 2017 | 2018 | 2019 |
|---|---|---|---|---|
| 22 | Caraș-Severin | 7478 | 7866 | 8873 |
| 30 | Hunedoara | 7550 | 8328 | 9564 |
| 34 | Arad | 9782 | 10,830 | 12,114 |
| 41 | Timis | 12,209 | 13,707 | 15,344 |

Source: INS (consulted in November 2022) [45].

**Table 9.** Turnover of county companies.

| County | Total Mill. EUR | Advantage Mill. EUR | CA/Employee EUR | Advantage EUR/Employee |
|---|---|---|---|---|
| Timis | 16,335 | 1221 | 76,695 | 5730 |
| Arad | 6814 | 465 | 75,994 | 5166 |
| Hunedoara | 771 | 220 | 44,089 | 3507 |
| Caras-Severin | 1442 | 116 | 47,183 | 3786 |

Source: INS (consulted in November 2022) [45].

*2.2. Methods Used in Researching the Current Level of Tourism Development in Caras-Severin County and in Romania*

To achieve the proposed research objectives, the case study method was used [46,47].

In order to research the current development level of tourism activity in Caras-Severin County and in Romania, and the dynamics of its development, so as to increase its contribution to the GDP of Romania and that of the county, we processed the dynamic statistical series by analytically adjusting the following indicators:

1. The number of tourist arrivals;
2. The leisure stays (days);
3. The annual revenues from tourism (million RON);
4. Share of tourism in GDP (%).

Using regression functions of the form Y = f(t), where the independent variable is t, the time interval of the dynamic series was calculated for a period of ten years (2011–2019), up until the emergence and development of the pandemic (2020–2021).

We used the set of mathematical forms of regression functions found and recommended in the specialized literature (with application in economic forecasting), as follows:

- The linear function y = at (1).
- Power function y = atx (2).
- The logarithmic function y = alog t (3).
- The second-degree polynomial function y = rt2 ± bt ± c (4).
- Higher degree polynomial function y = a1tn ± b2tn-1... ± an-1t ± an (5).

where (ai, i = 1,2,..., n), b, c are the coefficients of the regression function resulting from the processing of the dynamic series (analytical adjustment), and *t* is the time horizon; in the investigated case, t = 10 years (2011–2019).

## 3. Results and Discussion

### 3.1. Analysis of Tourist Activities in Romania and in the Banat Mountain (Caraș-Severin County)

From the comparative data (data before the pandemic, 2017–2018–2019) for the South-Eastern European countries (Romania, Bulgaria, Croatia, Slovenia and Serbia), it is found that the general tourist activity (qualitative and quantitative) of Romania is far below the level of the competing neighboring countries, as well as below the natural potential of our country. Although all three EU member countries (Bulgaria, Croatia and Slovenia) have a much smaller surface area and population, and the GDP per inhabitant is below the level of Romania (except in Slovenia), they have double the tourist intensity (in Bulgaria and Serbia), or 3.5 times greater in Slovenia and 6.4 times greater in Croatia. Even Serbia, a non-EU country, left without access to the sea (coastline) after the dissolution of Yugoslavia, has indicators of tourist activity far above the level of Romania.

In a county comparison, according to the data provided by the INS (2017–2019 average) regarding both total tourist arrivals and total foreign tourist arrivals (Figures 1 and 2), there is an extremely large disparity between counties at the national level. (Figures 1–3).

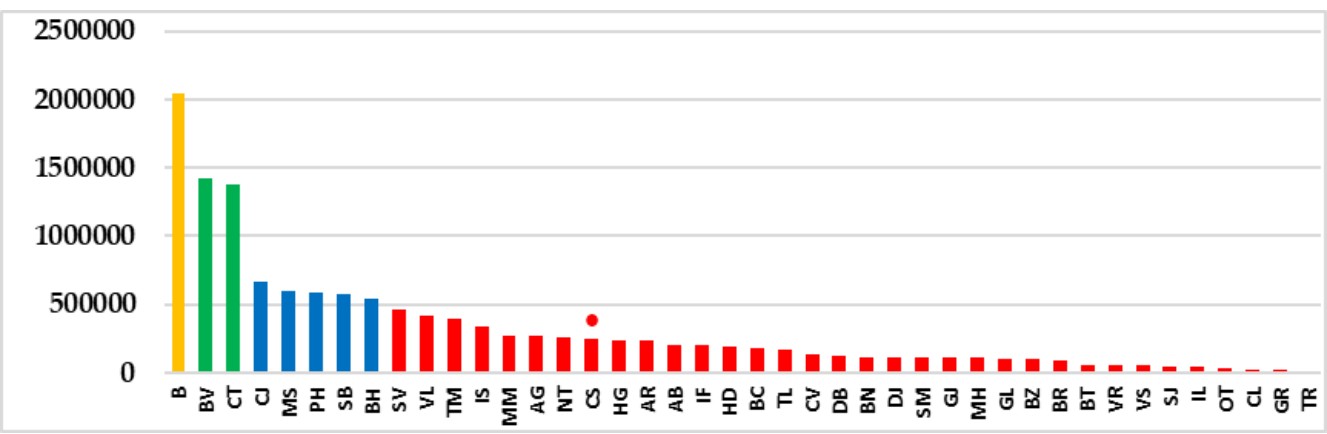

**Figure 1.** Number of tourist arrivals by counties in 2019. Source: INS (consulted in November 2022).

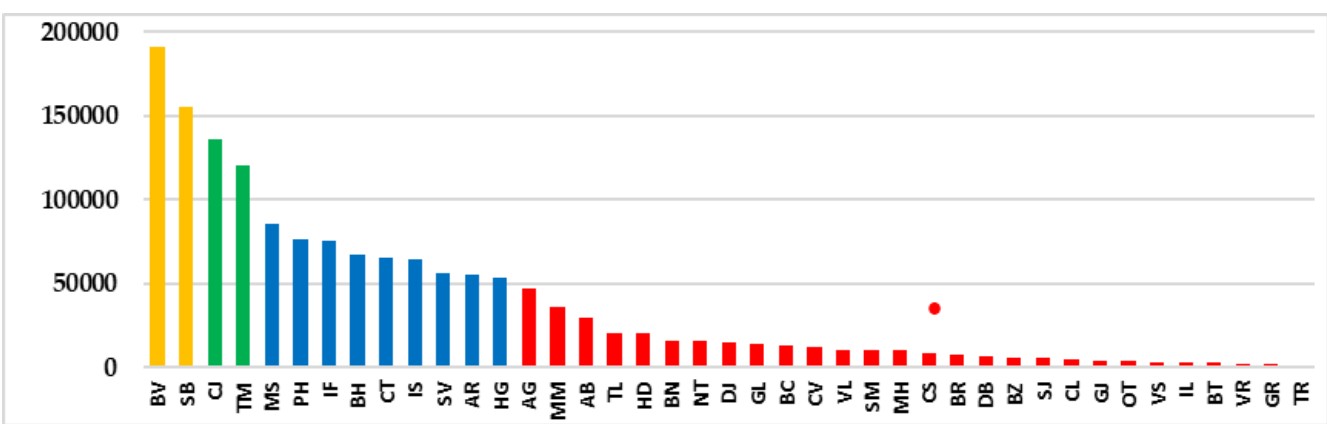

**Figure 2.** Number of foreign tourist arrivals by counties in 2019. Source: INS (consulted in November 2022).

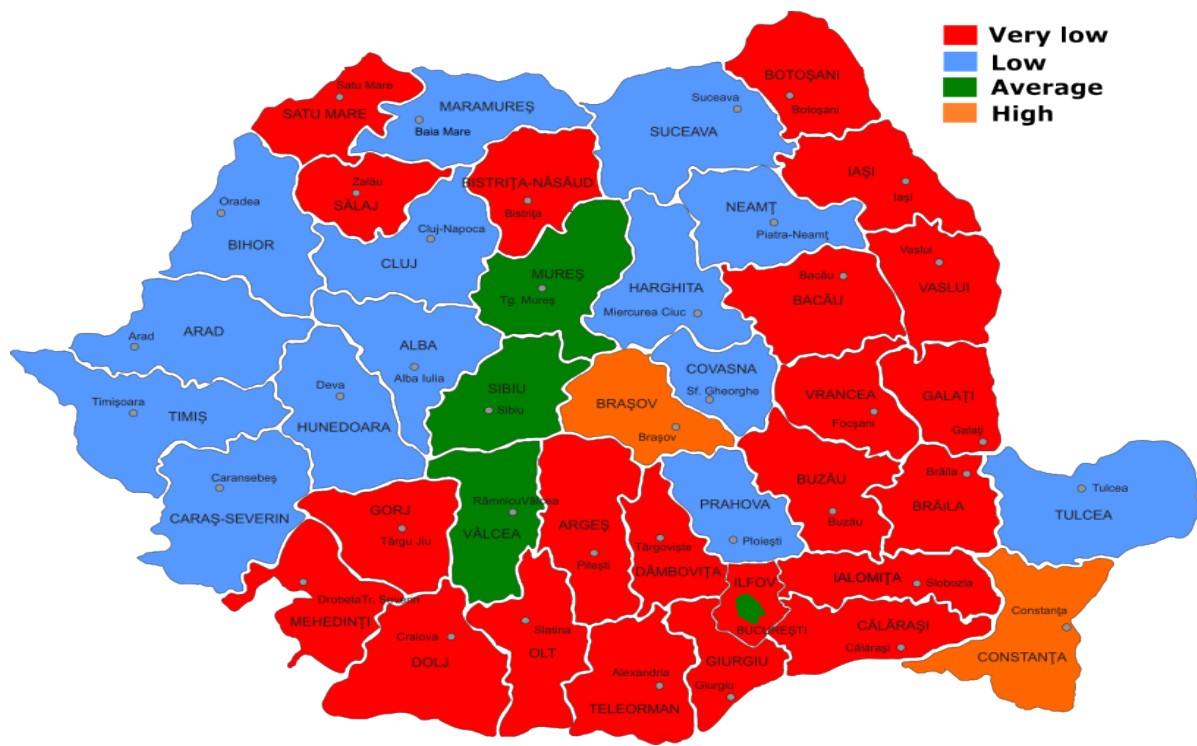

**Figure 3.** Map of tourism development by Romanian counties (tourist arrivals per 100 inhabitants). Source: own interpretation.

Based on the data regarding tourism intensity presented previously, and using the geographical map of the country, we have created a map of regional tourism development in Romania (Figure 3), as follows:

The Danube area (with the exception of Caras-Severin and Tulcea counties) has extremely low tourism in the counties of Teleorman (3.4 tourists per 100 inhabitants), Olt, Calarasi, Giurgiu, Ialomita, Dolj, Galati and Braila, with the average of the area being below 15 tourists per 100 inhabitants. Although the tourist potential of the Romanian Danube (fishing, hunting, sailing, sports, beach, etc.) is great, due to the absence of some facilities necessary for comfortable tourism, and minimal national and international tourism promotion, the general state of tourist use is far below the potential.

In the Moldova Region (Vaslui 13.7; Botoșani 13.7; Vrancea and Bacău, including Iași, 40.0) tourism has a low intensity (the average of the area is 25 tourists/100 inhabitants, 40% of Romania's average), although tourist offerings are found in significant and varied

numbers in this area: vineyards, in Vrancea and Iasi; mountains, in Bacau (Ceahlau); and a university, in Iasi.

The Subcarpathian area of Muntenia, with the counties of Prahova, Valcea and Arges, has tourist activity well above the Romanian average.

The area of Dobrogea, including Tulcea, with the Danube Delta, and Constanta, with the Black Sea coast, are still important attractions, especially for Romanian tourists.

The area of Transylvania (Cluj, Bihor, Mures and, especially, Sibiu and Brasov) is the region with the most intense tourist activity.

Regarding the V West Region, the three counties of Hunedoara, Timis and Arad, with lower natural potential (except for Hunedoara County), are close to the country's average, while in Caras-Severin County, having exceptional natural tourist offerings, the tourist intensity is higher than the average for the region (90.1 tourists per 100 inhabitants). This figure of tourist intensity for Caras-Severin County gives the county the qualification of a tourist county with high potential, but it is still far below its natural potential. (Figures 4 and 5).

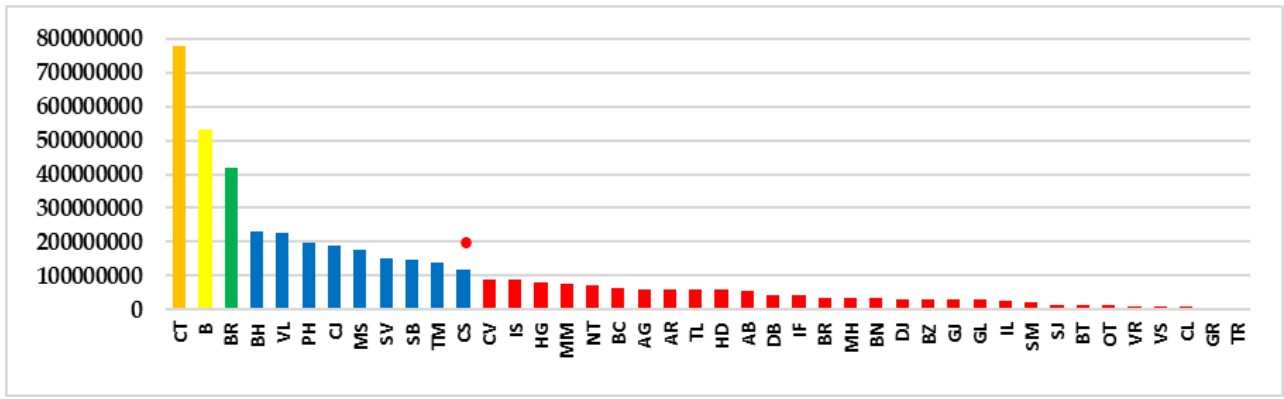

**Figure 4.** The value of revenues from tourism by counties in EUR. Source: INS (consulted in November 2022).

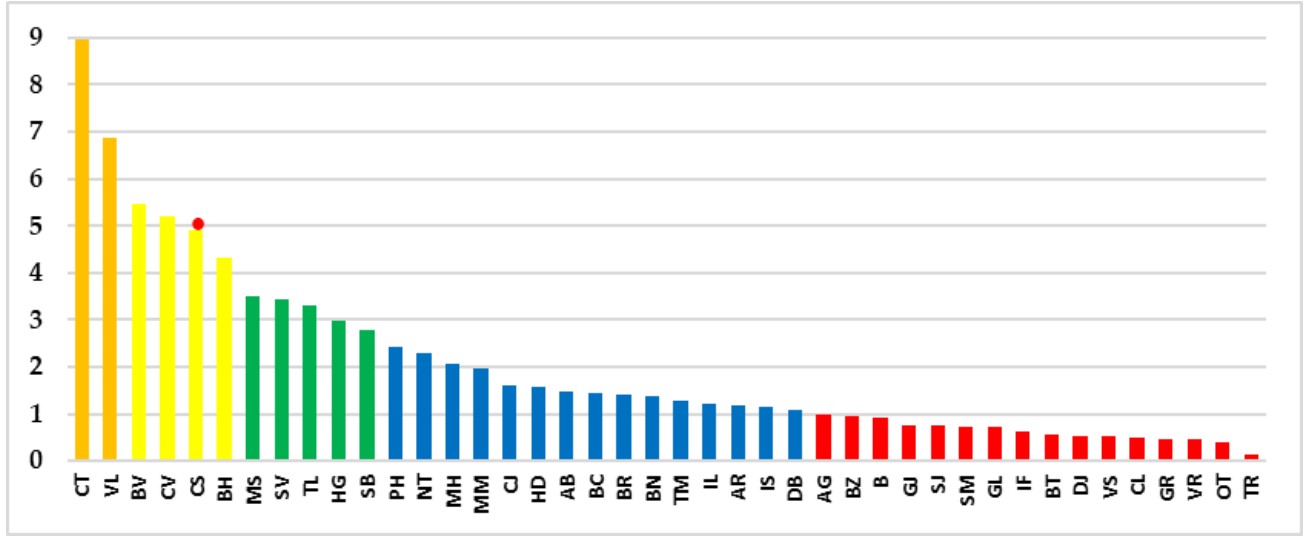

**Figure 5.** Share of revenues from tourism in GDP by counties. Source: INS (consulted in November 2022).

The indicators that allow us to appreciate the general state of Romanian tourism, including for Caras-Severin, are the intensity of tourism (number of tourists per 100 inhabitants) and the share of foreign tourists in the total arrivals, because neither the counties with

the most intensive tourism, nor any other Romanian tourist areas, come close to the numbers for the neighboring competing countries, Croatia and Slovenia. Even Bucharest—the capital of Romania—(111.2 tourists/100 inhabitants), where most of the business, scientific and diplomatic tourism is concentrated, has a tourist intensity below the average of Bulgaria (131) and even Serbia (128). The only counties with a higher tourist intensity, close to that of Slovenia and above the averages of Bulgaria and Serbia, are the counties of Constanta (204.9 tourists/100 inhabitants), from summer tourism on the Black Sea coast, predominantly for Romanian tourists, and Brasov (257.4), especially for winter tourism (skiing), but also for rest and hiking in the summer.

Regarding the indicator of the share of foreign tourists, Romania's situation is even more precarious (and dramatically so), for it is far below the level of competing neighboring countries (Table 10). In terms of foreign tourist arrivals, Romania has the worst results in the EU.

**Table 10.** Foreign tourists per 100 inhabitants.

| Country | 2017 | 2018 | 2019 |
|---|---|---|---|
| Romania | 13 | 19 | 19 |
| Bulgaria | 59 | 64 | 65 |
| Croatia | 381 | 408 | 433 |
| Slovenia | 177 | 202 | 234 |
| Serbia | 57 | 59 | 60 |
| Caras-Severin County | 4 | 4 | 4 |

Source: INS (consulted in November 2022).

From these indicators emerges the conclusion that the main factors of tourism attractiveness are not only the natural potential of the country, but also the ways in which the requirements of efficient and comfortable tourism are ensured: the tourism framework, hospitality and infrastructure, including the financial factor. It is enough to present, as an argument which confirms the previous statement, that the counties considered as the most attractive for tourists, namely, Brasov, Constanta, Sibiu, Bihor, Valcea, Suceava, Maramures, and including Caras-Severin County, register a very small number of arrivals of foreign tourists. Moreover, if we refer to the share of foreign tourists in the total tourist arrivals, the figures for these counties are unworthy to qualify them as the most important tourist counties of the country (Brasov 13.4%, Prahova and Maramures 13.1%, Tulcea 12.3%, Suceava 12%, Constanta 4.8% and Caras-Severin 3.4%).

The greatest shares of foreign tourist arrivals are registered in the counties (and cities) with strong potential from an economic, administrative and university point of view, where, in fact, the main forms of tourism are business, scientific, university and diplomatic tourism (as in the case of Bucharest). This category includes the following counties: Iasi (18.3%), Cluj (20.3%), Timis (30.4%) and, obviously, the country's capital, Bucharest (56.3%).

In summary, analyzing the data regarding tourist arrivals in Romania (total and total foreign), the revenues resulting from the number of overnight stays and the contribution of tourism to the GDP, the following conclusions can be drawn regarding the current stage of Romanian tourism, in general, and for mountain tourism in Caras-Severin:

— The way in which current Romanian tourism is conceived (strategically) and its development is totally inappropriate, compared to the requirements of a modern, efficient and extensive tourism and to Romania's exceptional natural tourist offerings. The fact that Romania has less than 1% of its GDP resulting from tourism is the most compelling figure demonstrating the "tourism precarity" of our country.

— On a national scale, on the country map, the development level of tourism is extremely dispersed. From the approximately EUR 4.5 billion in revenues from tourism in Romania in 2019, more than half was concentrated in five counties (Constanta, Brasov,

Bihor, Valcea and Prahova) and the city of Bucharest. Each of these five counties has at least one area (point) of important tourist attraction, such as the coast in Constanta, Baile Felix in Bihor, the ski areas in Brasov, Olt Valley and the monasteries in Valcea and Prahova Valley in Prahova.

— The country's tourist counties (Maramures, Suceava, Tulcea, Sibiu and Caras-Severin), although they have exceptional natural (and to some extent also anthropic) tourist offerings, contribute only 17.1%, or RON 551 million (approx. EUR 110.2 million), to Romania's tourism revenues. Although in these counties the tourist offerings are substantial (tradition in Maramures; the monasteries and Dornelor land in Suceava–Bucovina; the Danube Delta in Tulcea, and the European cultural capital Sibiu; the Herculane Baths and the Danube Gorge in Caras-Severin), their utilization is still far below their potential.

— There are eight counties in Romania (a quarter of the country's counties) where tourism is practically non-existent from an economic point of view (Teleorman, Giurgiu, Calarasi, Vaslui, Vrancea, Olt, Botoșani and Salaj), with annual revenues from tourism activity being under RON 20 million (approx. EUR 4 million) and under 100,000 annual overnight stays in each county.

— At the current level of tourism in Caras-Severin County (relatively good numerically, 90.1 tourists/100 inhabitants), it is in the 10th place in the top counties, but an important remark that must be made is regarding the distribution of tourists by tourist destinations. Of the 244.6 thousand tourists registered in the period before the pandemic in Caras-Severin County, over 50% had as their destination the Herculane Baths resort, and 26% the Semenic and Muntele Mic ski areas, and only about 20% were there for other forms of tourism practiced in the mountain area of Banat.

— Alarming for Caras-Severin County is the fact that rural tourism and agritourism are practically non-existent. Compared to Suceava and Maramures counties, agritourism in Caras-Severin is very poorly represented. While in Caras-Severin County the accommodation capacity in rural guesthouses and agritourist guesthouses represents 22% of the total county capacity, in Suceava county, for example, it is double that at 41%. In Suceava county, there are six communes (Dorna Candrenilor, Humor Monastery, Gura Humorului, Scheia, Sucevita and Vama) with over 300 accommodation places in the commune, while in Caras-Severin County there are only three communes (Valiug, Brebu Nou-Garana and Poiana Marului) falling into this category. However, in the communes of Suceava County, most of the rural guesthouses are agritourist guesthouses, while in Caras-Severin County, in this case in the communes of Valiug, Brebu Nou—Garana and Poiana Marului, there is no agritourist guesthouse, and there tourism is focused on the specific activity of "vacation villages".

Furthermore, the development of human resources for tourism must be a priority in order to offer the quality services expected by the tourism market, and this requires a specific, systematic approach to projecting staffing needs and establishing the training modalities needed to provide qualified staff in both the public and private sectors.

### 3.2. Identifying the Tourism Forms Practiced in the Mountain Area of Banat (Caras-Severin County)

The forms of tourism that can be practiced in Caraș-Severin County are: (Table 11).

**Table 11.** Forms of cultural tourism practicable in Caras-Severin County.

| | | |
| --- | --- | --- |
| Cultural–historical tourism | Consists of visiting historical monuments and ensembles, as well as archaeological sites in Caras-Severin County | Tibiscum Camp in Jupa-Caransebes, Roman baths in the Herculane Baths, Tabula Traiana in Cazanele Dunarii, the remains of the Villa Rustica in Gornea, Sichevita commune, etc. |

**Table 11.** *Cont.*

| Cultural–architectural tourism | Aims to visit some monuments and architectural ensembles in Caras-Severin County | The "Mihai Eminescu" Old Theater building in Oravita, the Cultural Palace in Resita, the Palace of the Community of Wealth in Caransebes, the Baroque-style historical center of Herculane Baths resort, the Baroque-style train station in Herculane Baths, the Oravița train station, the Caransebes Town Hall building, etc. |
|---|---|---|
| Cultural–museum tourism | Means visiting historical, ethnographic, art, archaeology, natural sciences, science and technology, memorials and village museums in Caras-Severin County | The Mountain Banatul Museum in Resita, the County Museum of Ethnography and of the Border Regiment in Caransebes, the Station Museum in Baile Herculane, the Open Air Steam Locomotive Museum in Resita, the "Constantin Gruescu" Iron Mineralogy Collection in Resita, the Museum of the Amateur Cinematographer, and the village museums in Bania, Cornea, Gornea, Mehadica and Racasdia. |
| Cultural–scientific tourism | Involves exploration and scientific discovery in Caras-Severin County | Aiming at the past (visiting archaeological sites), the present (visiting active enterprises and industrial parks) and the future (visiting the UBB Cluj Research Center–University Campus of Summer in Coronini/Resita). |
| Cultural–religious tourism | Aims at pilgrimage and visiting monasteries, hermitages, churches and cathedrals in Caras-Severin County and the Orsova area | Calugara Monastery in Ciclova Montana, Piatra Scrisa Monastery in Armenis, Vasiova-Izvor Monastery in Bocsa, Teius Monastery in Caransebes, Saint Ana Monastery in Orsova, Mraconia Monastery in Dubova, Brebu Monastery in Brebu-Soceni, Slatina Nera Monastery in Sasca Montana, Almaj-Putna Monastery in Putna-Prigor, Bazias Monastery, " St. Elijah " Hermitage in Mount Semenic, Poiana Mărului Hermitage, the translated Orthodox Church in Resita, the "Immaculate Conception" Catholic Church in Orsova, the Gothic-style synagogue in Caransebes, the Episcopal Cathedral in Caransebes, the collection of religious art in the Caransebes Bishopric Building. |
| Cultural–industrial tourism | Is linked to visiting many unique industrial and technical sites in the mountain area of Banat | The first railway in Romania at Oravita–Bazias, the Oravita viaduct, the first mountain railway in Romania at Oravita–Anina, the open-air museum of steam locomotives in Resita, artificial lakes in Barzava and Timis (Secu, Breazova-Valiug, Gozna-Crivaia, Trei Ape), first well of the deepest mine (1000 m) in Europe in Anina, marble quarry in Ruschita, the water mills with a horizontal wheel at Valea Rudariei, in Moceris and Sichevita, the Furnace and the Funicular in Resita, etc. |
| Cultural–ethnographic and folkloric tourism | Can be practiced on the occasion of original and authentic folklore events in Caras-Severin County | The "Hercules" International Folklore Festival at the Herculane baths resort, the Almaju Country Festival, the Gugulani Country Festival, as well as the "nedei" (prayers) which, from spring until late autumn, are held in all Banat villages with Orthodox churches that have patron saints, with specific religious and secular traditions and customs. |
| Cultural–artistic tourism | Focused on festivals and cultural–artistic events throughout the year in Caras-Severin County | Jazz Festival in Garana, "Mihai Eminescu" Days in Oravita, the "Crystal Palette" painting colony in Garana, the sculpture camp in Teius park in Caransebes, the "Decade of German Culture" event in Resita, the gastronomic festival "The Golden Cauldron" in Moldova Noua, etc. To these is added the "Days of Culture" in each city of Caras-Severin County. |
| Cultural–ethnic tourism | Can be practiced through the multiculturality of mountain area of Banat on tourist routes specific to each ethnic folklore | The German folklore route, the Croatian folklore route, the Serbian folklore route, the Czech folklore route, the Ukrainian folklore route, the Hungarian folklore route, etc. |
| Cultural–itinerant tourism | Can be carried out in the form of thematic circuits through several localities in Caras-Severin County | The Iron Way or the Road of the Romans, etc. |

Source: own interpretation.

I.   Mountain tourism

This is the main form of tourism in Caras-Severin County for those who love hiking, seek rest and recreation (relaxation), prefer fishing and hunting, try mountaineering and, above all, are passionate about recreational or sports skiing in the Valiug mountain resorts—Semenic—Garana and Muntele Mic—Nedeia (Tarcu). Mountain tourism is increasingly assimilated to winter sports tourism. If, in the past, the motivation "to climb the mountain" was to benefit from the climatic conditions beneficial for rest and treatment, hiking and climatic cure, today the number of tourists seeking out the mountain to ski has increased exponentially, so that the practice of winter sports tends to become the main motivation for mountain tourism.

A motivation, which is not new but has been rediscovered by tourists, is the return to nature, by preferring to travel to a "natural place", or a place little transformed (anthropized) by human hands. Many tourists choose mountain destinations where they feel they can recreate in perfect harmony with the surroundings. Some new tourism practices for spending free time in the mountains that have more and more participants are paragliding, rafting or descending by boat on swirling waters, climbing, horseback riding, fishing, birdwatching (looking and listening to birds), etc. Mountain tourism in Caras-Severin County can be grouped into several new forms of tourism that can be practiced on the mountain: mountain hiking tourism, rest and recreation tourism (relaxation), hunting and sport fishing tourism, climbing tourism (mountaineering), speotourism, cycle tourism and ecotourism.

II.  Spa tourism

This is the second form of tourism practiced in Caras-Severin County, which combines relaxation with various forms of cure and treatment. The main resort (the only one in Caras-Severin County) regarded as being for spa tourism is the Herculane Baths spa resort, due to the multiple qualities of its 20 thermal springs. If, in the past, the Băile Herculane resort included treatment courses (curative spa tourism), today the spa product offered includes beautifying and revitalizing procedures and treatments (wellness tourism) or SPA—health through water. The term SPA comes from the Latin language through the expression "sanitas per aquam", which means health through water. Wellness tourism, through SPA centers, is the most promising form of niche tourism. In the Herculane Baths resort, visitors can also practice hiking tourism, leisure tourism, mountain tourism, speotourism, meetings and adventure tourism, so the resort offers diversified and complementary forms of tourism.

III. Cultural Tourism

This is the form of tourism that makes the cultural attractions the center of the offering. In Caras-Severin, cultural tourism is distinguished by different (sub) forms, depending on the main categories of cultural tourist objectives. Each form of cultural tourism consists of cultural assets from the cultural heritage and cultural events:

IV.  Agritourism

Agritourism in most of the mountainous areas in Romania, including in the mountain area of Banat, is a potentiality rather than a reality. The expansion and generalization of agritourism requires a deep remodeling of the rural infrastructure and the equipment of the agritourism households (farms), to make it suitable for tourism. In addition to equipping the agritourism households (farms) and improving the rural infrastructure, for the expansion of agritourism, a promotional tourism management is also necessary, including the establishment of tourist information networks through which the supply of agritourism can be brought as close as possible to the demand for agritourism, and the promotion of this type of tourism as an educational tourism for students (and even residents) from the cities, who do not know enough about "country life" and the activities characteristic of agriculture, fruit growing, raising milk cows, shepherding, rural customs and traditions, etc.

Permanent residence (habitat) in the Carpathians, including in the mountain area of Banat, rises up to a maximum altitude of 1000–1400 m, while in the Swiss or Austrian Alps the permanent residence (or quasi-permanent, compared to the mountain habitat of Romania) rises almost 1000 m higher, meaning up to 2000–2200 m, there being big differences between the mountain economies of Romania and other mountain countries in the EU. While in Switzerland and Austria, to use the same examples, the government policies for the mountain economy of these countries support the expansion and consolidation of the mountain habitat and the economy of mountain towns, in Romania, the mountain economy, in general, and that of Caras-Severin County, in particular, is in dramatic decline (with two exceptions: the expansion of forest exploitation in accessible forests, and the expansion of holiday home areas or the transformation of depopulated villages into holiday villages and habitat in mountain villages) Why? What are the causes?

V.   Ethno-Cultural Tourism

The main elements of the ethno-cultural tourism potential of Caras-Severin are the following (Table 12).

**Table 12.** The main elements of the ethno-cultural tourism potential of Caras-Severin.

| | | |
|---|---|---|
| The rural settlement of mountain towns in Banat, the construction and decoration of the peasant houses | In the mountain area there are valley towns, located on the edge of a river and along the roads | Cornereva is the largest locality in Romania, with 36 hamlets, the most famous of which is Inelet because of the access stairs in the hamlet, followed by Sichevita with 19 hamlets |
| Traditional technical installations | A special place is occupied by the "water mills" with a horizontal wheel (with bucket) | In Rudăria (gutter and bucket mills) and Sichevita, Moceris, Sopotu Vechi and Valea Rosie-Sopotu Nou (bucket and cufflink mills) |
| The folk costume | This is a true "visiting card" of Banat villages, being a clothing ensemble with distinct pieces of clothing and ornaments, of great artistic value | Costumes of the Gugulani, Almajeni and Carasovani; a particular attraction is the specific holiday costume, which includes many decorative and chromatic elements |
| The folk art of woodcraft | The use of wood and its processing in various forms was done according to the needs of the household and the artistic sense of the folk craftsmen | The "wooden cradle" for carrying children on the back can rightly be considered one of the most interesting and beautiful creations of the craftsmen in the Nera Valley and the Danube Gorge |
| The popular art of marble processing | The existence of the marble quarry favored the artistic processing of marble and the presence of some elements in the construction of houses (pillars, stairs, marble tables, floors) and funerary pieces | In Ruschita |
| The popular art of metalworking | The steel and cast-iron embroideries | That on the bridges over the Cerna, at Băile Herculane and in the construction of house fences in Rusca Montană (craftsman Ilie Nicoară), as well as funerary pieces, stand out |
| The folk art of leather processing | The art of leather processing had an exceptional development until the 1980s–1990s | There are fewer and fewer craftsmen who deal with leather and shoemaking (Valisoara, Cornea, Cuptoare) and who make leather shoes, jackets, hats, all impressive in their decoration and style |
| The art of fabrics, stitches and embroidery | Popular pieces | Sewn by the skillful hands of the Banat woman, constitutes a true Carasan folk art |
| The art of pottery | From the Roman ceramics | In Binis |
| The art of braiding | Various hand-woven objects | In Carasova |

Source: own interpretation.

VI.   Active Tourism in Protected Areas

Given the number and size of protected areas, active tourism in protected areas is one of the forms of ecological tourism proposed by the strategy for balanced development in Caras-Severin County. This form of tourism "welds" perhaps best the relationship between

the tourist and the place visited, through direct interaction. Tourists seek to rediscover themselves while climbing a cliff, passing through cave galleries, or observing birds and animals, as is now the case at Măgura Zimbrilor near Armeniș. This is how active tourism was born and will develop in the national parks and protected areas of Caras-Severin county, a form of ecological tourism with a deeper and more comprehensive meaning, which aims to achieve a double, balanced objective: on the one hand, to protect nature from the negative influences of man and, on the other hand, to preserve the same natural environment for the benefit of man, for rest and recreation. Visitors of all ages are and will always be delighted to discover the mysteries of the county's outstanding nature parks. There are four national parks in Caras-Severin County, and the fifth, Retezat National Park, includes a small alpine area in the territory of Caras-Severin County. They are Semenic National Park—Cheile Carașului; Cheile Nerei—Beușnița National Park; Domogled—Valea Cernei National Park; Porțile de Fier Natural Park; and Retezat National Park, bordering Caras-Severin County. From the most compact virgin forest in Europe, the Izvoarele Nerei beech forest, to the newest microdelta in Europe formed at the Nerei's spill into the Danube, and from the longest sector of gorges in Romania to the longest gorge in Europe, Caras-Severin County offers wild landscapes for adventure lovers, but also picturesque landscapes for those seeking active relaxation in nature. National parks are the best places for tourism, environmental education, environmental protection, school and student activities and scientific tourism.

### 3.3. Comparative Forecasts of Tourism Development in Caras-Severin, Suceava and Bihor Counties and in Romania

For comparison with the tourism situation in Caras-Severin County, we have also studied, adjusting the regression functions, two other counties with relatively similar natural and anthropic conditions, namely, the counties of Bihor and Suceava. The regression functions were also calculated for Romania's tourism.

After testing several forms of the regression functions (1,2,...,5), taken with the Excel program, with the help of correlation coefficients, the best adjustments resulted in the regression functions presented below. Graphical representations of the regression functions (Y1, Y2, Y3 and Y4) for the size of the indicators (the number of tourist arrivals, the leisure stay, the amount of the receipts from tourism and the share of tourism in the GDP) are presented in Figures 6–9.

**(1) The number of tourist arrivals, y1—polynomial function of the third degree**.

| Caras-Severin | $Y_{1C} = -0.1411x^3 + 3.5295x^2 - 6.5609x + 99.912$ |
|---|---|
| Suceava | $Y_{1S} = -0.0563x^3 + 3.2773x^2 - 0.1716x + 201.84$ |
| Bihor | $Y_{19} = -0.5409x^3 + 11.411x^2 - 26.453x + 226.31$ |
| Romania | $Y_{1R} = -8.5158x^3 + 164.53x^2 - 65.306x + 6255.5$ |

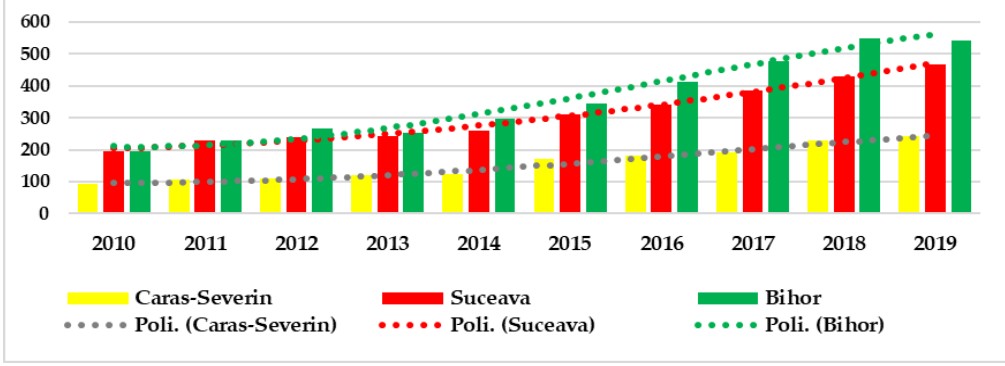

**Figure 6.** The number of tourist arrivals in Caras-Severin, Suceava and Bihor counties, thousands. Source: own interpretation.

**Function $Y_1$–the number of tourist arrivals** in the studied counties and in Romania, both come from the analytical interpretation of the function [Y = f(t)] and from the graphic interpretation; being strictly increasing, from an economic point of view it describes a positive tourist reality for the past period (2010–2019), as well as for the future forecast, a fact confirmed by the partial data of tourist arrivals from 2022 (post-pandemic year). On the other hand, significant differences are found between the three counties, with Caras-Severin County being far below what we might expect from its exceptional natural and anthropic tourist offerings. The explanation for the much lower number of arrivals registered in Caras-Severin can only be the much reduced, unprofessional and poor presentation of tourism for this county, especially in terms of promoting the Herculane Baths resort and the exceptional tourist area the Danube Gorge.

**(2) Length of stay 2000-2019, y2–polynomial function of the third degree.**

| | |
|---|---|
| Suceava | $Y_{2S} = 0.0026x^3 - 0.043x^2 + 0.1618x + 2.2543$ |
| Bihor | $Y_{2B} = 0.0012x^3 - 0.0071x^2 - 0.2558x + 4.823$ |
| Caras-Severin | $Y_{2C} = 0.0006x^3 - 0.0103x^2 - 0.1748x + 5.308$ |
| Romania | $Y_{2R} = 0.0004x^3 - 0.0028x^2 - 0.0539x + 2.6843$ |

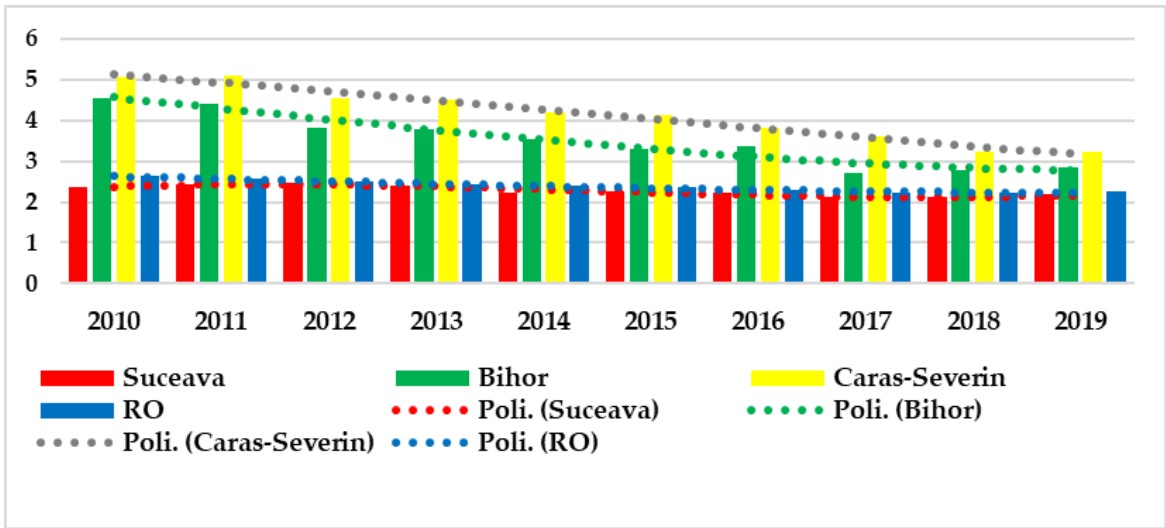

**Figure 7.** The length of leisure stay in Caras-Severin, Suceava and Bihor counties and in Romania, days. Source: own interpretation.

**The interpretation of the function $Y_2$–length of leisure stay**, a polynomial function of the third degree, has an obviously decreasing trend for Caras-Severin and Bihor counties, from a length of leisure stay of 4.5–5 days in 2010, to 3–3.5 days in 2019. The longer length at the beginning of the studied period (2010) is explained by the greater share of tourist stays in Romania's two important spa resorts, the Herculane Baths in Caras-Severin and the Felix Baths in Bihor. Over time, after 2010, both counties diversified their tourist offerings, in the sense of increasing shorter tourist stays, usually on weekends in the mountain areas of Banat and the Danube Gorge, in Caras-Severin or in the mountain areas of Bihor County.

**(3) The value of receipts from tourism, y3–polynomial function of the second degree.**

| | |
|---|---|
| Suceava | $Y_{3S} = 12.473x^2 - 31.156x + 337.6$ |
| Caras-Severin | $Y_{3C} = 6.3519x^2 + 6.1529x + 269.08$ |
| Bihor | $Y_{3B} = 15.97x^2 - 18.739x + 549.54$ |
| Romania | $Y_{3R} = 258.36x^2 + 211.12x + 9397.5$ |

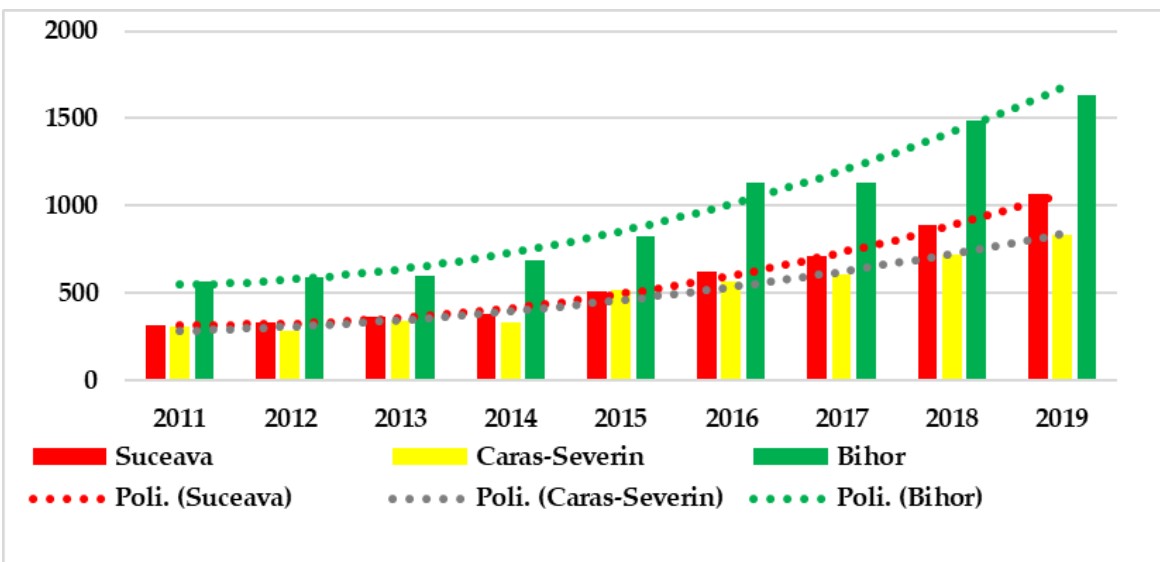

**Figure 8.** The value of receipts from tourism in Caras-Severin, Suceava and Bihor counties, thousands RON. Source: own interpretation.

**Interpretation of function Y₃–share of tourism receipts**, a polynomial function of the second degree. The graphical representation of the Y3i functions in Figures 7 and 8 highlights, both in the case of the counties and for Romania, the increase in the value of receipts from tourism activities, explained by the increase in the number of tourists and in the price of a tourist day.

**(4) The share of tourism in GDP, y4–polynomial function of third degree.**

| | |
|---|---|
| Suceava | $Y_{4S} = -0.0119x^3 + 0.1957x^2 - 0.6098x + 3.4367$ |
| Bihor | $Y_{4B} = -0.0207x^3 + 0.3318x^2 - 1.1903x + 5.5137$ |
| Caras-Severin | $Y_{4C} = -0.0169x^3 + 0.2672x^2 - 0.8071x + 5.2814$ |
| Romania | $Y_{4R} = -0.0082x^3 + 0.1256x^2 - 0.3688x + 2.101$ |

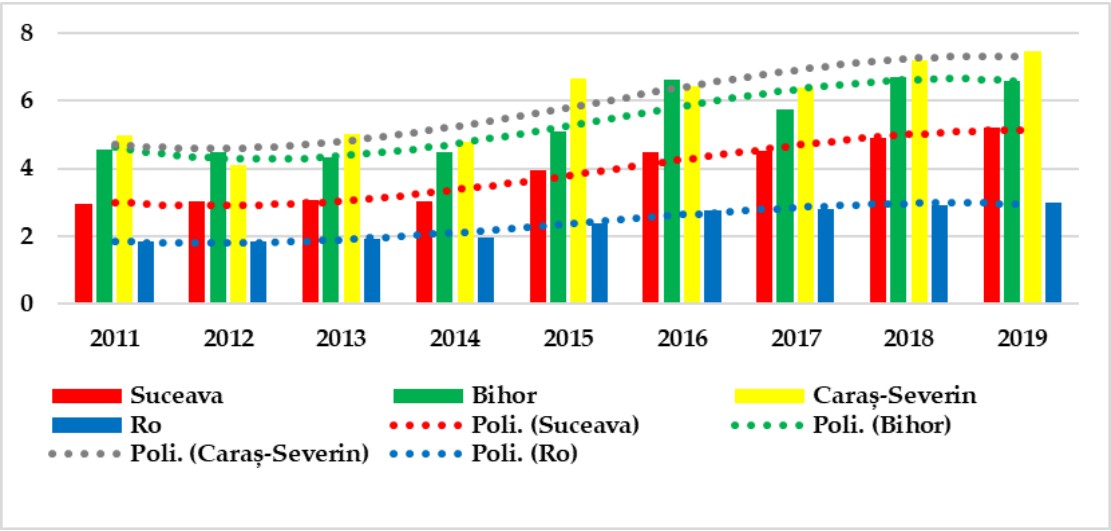

**Figure 9.** The share of tourism in GDP in Caras-Severin, Suceava and Bihor counties and in Romania, %. Source: own interpretation.

**Interpretation of function Y₄–share of tourism in GDP**, a polynomial function of the third degree. From the graphic representation (Figure 9), it can be seen that the highest share of tourism in GDP is registered in Caras-Severin County (5–7%), and the lowest share is found in the case of Romania, albeit increasing in size from year 2011, at 1.9%, to 3% in 2019. Compared to competing neighboring countries, Croatia (6%) and Slovenia (3.5–4%), Romania still, among European countries, has an insufficient contribution of tourism to its GDP. The explanation for this still too low share of tourism in its GDP, and for the slow growth trend, is that it results from the excessively large discrepancies in the intensity of tourism and receipts from tourism among the country's counties. With the exception of five or six counties (Constanta, Brasov, Valcea, Caras-Severin, Bihor and Suceava), in most counties the receipts from tourism are below 2–25% of GDP.

Romania can approach a 6% share for tourism (as much as Croatia currently achieves) only if it accelerates the average annual growth rate of receipts by improving, from all points of view, its tourist offerings.

With the current growth rate of the share of tourism in GDP of 0.11% [(3 − 1.9)/10 = 0.11; 1.9% GDP 2010; 3% GDP 2029], Romania can reach 6% (equal to Croatia, 2019) in 27.2 years (3:0.11 = 27.2 years). Simulating the average annual growth of tourism in the GDP, by a doubling, tripling, etc., of it, Romania can reach 6% after 27 years by maintaining the current average annual rate (0.11%/year), or after a decreased number of years, depending on the average annual growth of the share. Consider it reasonable to triple the annual rate (from 0.11%/year to 0.33%/year), which would lead to reaching a 6% share of tourism in Romania's GDP within 9 years (Figure 10).

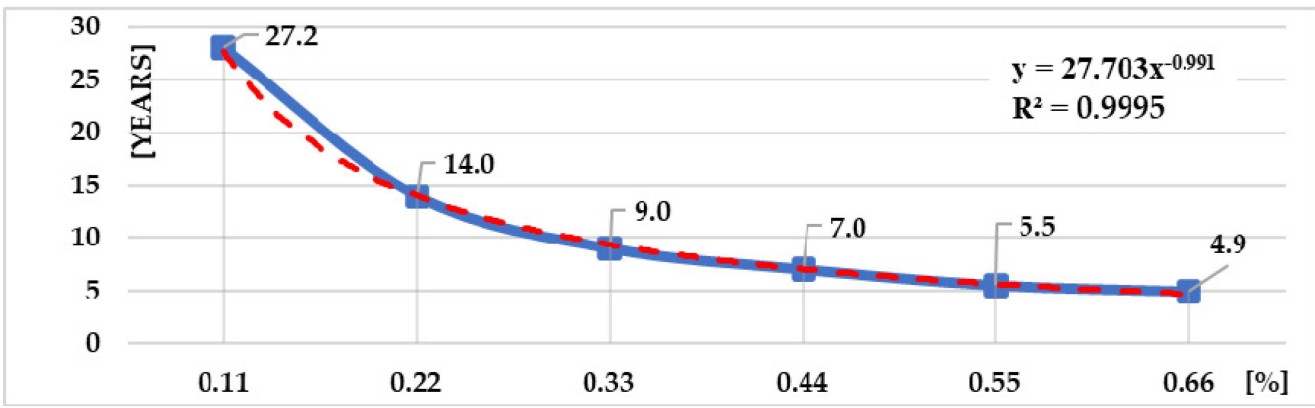

**Figure 10.** Time required (years) to reach a 6% share of tourism in Romania's GDP, according to the annual increase in receipts from tourism (%). Source: own interpretation. Using the Smoothed Line function, the evolution trend of the data series was plotted in blue. Moreover, the power regression function was also estimated, marked on the diagram by the red dotted line, together with its mathematical equation, respectively the value of the coefficient of determination $R^2$.

## 4. Conclusions

For Romania, sustainable economic growth means directing investments and funds towards sustainable projects and activities.

In addition to generating economic growth and stimulating the development of local communities, sustainable rural tourism, especially agritourism, represents travel opportunities for low-income families, due to the fact that this type of tourism does not charge very high taxes. This means that, from a social point of view, it creates a sense of inclusion for lower-income social groups by offering the opportunity to enjoy the beauty of the environment, natural and man-made attractions, and the traditional agri-food products of the rural areas concerned [48].

Caras-Severin County, like the entire Banat region, is unique in the Romanian and European space, being the place where an interethnic, multicultural and interfaith model was created in the true sense of the word.

The 15 main communities of Romanians, Germans, Hungarians, Croats, Serbs, Czechs, Slovaks, Ukrainians, Lipovian Russians, Turks, Bulgarians, Jews, Poles and Italians make in the Romanian Banat a region of European culture, considered a "Little Europe", and a model of peaceful interethnic coexistence.

If, in other places, the ethnic communities are in competition, which is not always understood in a positive, constructive, harmonious sense in Caras-Severin County, each ethnic group has managed to preserve its language and develop its culture. Moreover, the citizens belonging to the various ethnicities seek to know and speak the language of each of the other communities and to know well the cultural values of the others.

If Romanian tourism maintains its current growth each year, it will reach a 6% share in the country's GDP within 27 years. This is a concerningly slow development rate, compared to other economic branches, once again highlighting the low level of interest shown by the government in this branch. In order to reduce this time span, I recommend that the Romanian government take the following measures: create a specialized Romanian Tourism Promotion Authority that will oversee the general affairs of tourism within the country, and that will create a national tourism development strategy that will serve as an example for all the counties. This authority will also oversee the implementation of the proposed projects within the areas of interest, especially the underdeveloped counties with high potential, such as Caras-Severin and the Banat region. Moreover, the same office will restructure the entire network of tourism informational centers, creating a strong national brand and unity within tourism promotion materials. It would be preferable that this promotion authority be an outsourced service, rather than a governmental branch, due to a lack of qualified personnel in the field of tourism being hired within the government and the Ministry of Entrepreneurship and Tourism.

According to the authors, the most beautiful trips can be made as follows:

- On the water, on a cruise through the Danube Gorge from Moldova Noua to Orsova, through the most beautiful Danube Gorge, from the springs to the spill;
- By train, on the first mountain railway in Romania, Oravita-Anina in the Aurora Banatului tourist area and, at the same time, one of the most beautiful railways in operation in Europe;
- By car, to the Old "Mihai Eminescu" Theater in Oravita, the first theater in Romania, to the Rudaria Mills in the Aurora Banatului tourist area, and to the Rusca Montana Tourism Monument, unique in the world of tourism, in the Muntele Mic—Poiana Malului tourist area;
- By bicycle, on the mountain paths and former forest paths from Resita to the Comarnic Cave, in the Semenic tourist area;
- On foot, for a tour of the Herculane Baths resort, the oldest balneo-climatic resort in Romania and South-Eastern Europe, in the Nera-Beusnita Gorges, the longest and most beautiful gorges in Romania, in the Aurora Banatului tourist area and in the Carasului Gorges, the wildest in mountain Banat, in the Semenic tourist area.

In conclusion, the forms and tourist areas of the mountain Banat offer an extremely wide range of tourist activities that the tourist can engage with in Caras-Severin County, which are, simply, of a completely exceptional potential. The number one problem for tourism in this area is, however, the implementation, despite the value of this immense potential, which, we appreciate, is still too little used, as the present study has highlighted.

The information obtained as a result of the studies and research carried out in the field can be used by the decision makers of Caraș-Severin County for the realization of a unitary strategy for the development of tourism, based on the natural and human resources identified in the researched area, which is so necessary for the mountainous area of Caraș-Severin County. Furthermore, the results obtained can be the basis for the realization of joint programs and projects for the development of the area, thus contributing to the development of the local business environment and which, through their long-term consolidation, can generate positive economic and social externalities.

**Author Contributions:** Conceptualization, N.M.-S. and P.-D.P.-O.; methodology, investigation, data curation, writing—original draft preparation, writing—review and editing and supervision, N.M.-S. and P.-D.P.-O.; project administration, N.M.-S. and P.-D.P.-O. All authors have contributed to the study and writing of this research. All authors have read and agreed to the published version of the manuscript.

**Funding:** This research paper is supported by the project " Increasing the impact of excellence research on the capacity for innovation and technology transfer within USAMVB Timisoara" code 6PFE, submitted in the competition Program 1—Development of the national system of research—development, Subprogram 1.2—Institutional performance, Institutional development projects—Development projects of excellence in RDI.

**Institutional Review Board Statement:** Not applicable.

**Informed Consent Statement:** Not applicable.

**Data Availability Statement:** The data presented in this study/paper are available based on a request from the corresponding author.

**Conflicts of Interest:** The authors declare no conflict of interest.

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
