# Peer review of "Analyzing the Development Possibilities of the Mountain Area of Banat, Caras-Severin County"

_sustainability, doi:10.3390/su15118730_

Round 1

Reviewer 1 Report

The authors describe the economic but also the socio-cultural effects of tourism in Banat, specifically Caraș Serverin, effects that contribute to the development of the rural space, being an alternative to solve, at the same time, problems that concern, on the one hand, the village, and on the other hand the city.
We can say that the social effects of tourism development are not limited only to the population of the area where it takes place, but also have implications for the tourists who benefit from this activity. Thus, due to the fact that the rates charged in this form of tourism are more affordable, it offers the chance that social groups with a lower level of income can also benefit from the agricultural and tourist resources, from the atmosphere in the rural environment. At the same time, it ensures for tourists to spend some alternative vacations simultaneously with increasing the degree of knowledge and appreciation towards the traditional rural life.

The research needs to provide more background about the tourism topic in terms of its challenges, gaps, potentials and futures especially about the lack of proper data for design and planning. Scale of tourism needs be discussed and the effectiveness of the proposed method. The chosen method has no degree of novelty!

How can the research support agritourism against climate change impacts and support biodiversity and more resilient rural landscapes?

Author Response

Dear Reviewer,1

First of all thank you for time, spend to review our paper, then for your kind observation and proper suggestions. Your proper suggestions were beneficial to help us order/direct our scientific paper work.   So we re-organized our paper, more logically as you recommended us, in the following we will mention the main changes, highlighted with red color in the paper:

  • Concerning your remark about background for the tourism topic we have make some adings to sustain the benefits of this field, lines 145-151,55-59, 72-99;
  • As far as concerning the comment about the research support agritourism at lines 240-252 we did mention the importance of this field of tourism for Caras Severin area;
  • Regarding the impact over the environment was discussed at lines 113-127;
  • Regarding the support of agritourism against climate change impacts and support biodiversity and more resilient rural landscape, we did implement this suggestion in conclusion part, at lines 815-825 and 842-862;
  • As far as concerning the effectiveness of the proposed method we discussed it and justify it at lines 160-184.

So in the end we hope we've been able to implement all your requirements. Thank you for your valuable suggestions and the time dedicated to reviewing our paper. You help us to direct our study on a better and much improved line.

The authors

Reviewer 2 Report

Dear Authors,

Your paper is interesting but it needs to be revised.

I would suggest synthesizing the text between lines 54 and 70 avoiding to repeat 3 times “Minimizing the”:

I would suggest synthesizing the Paragraph III. Cultural tourism including in a single table the text between lines 445 and 511.  The list is too long to be read.

Paragraph V. Ethno-cultural tourism is also too long.

Moreover I suggest:

1. you are requested to better check the text to solve typos and problems related to spacing and punctuation (example line 364, 374).

2. Currency should always be reported in Euros. It is fine to express the values in leu, the Romanian currency, but it is advisable to always put the value in euros at least in brackets (eg lines 623, 625).

3. paragraph 3.2 (lines 408 and following) would need some bibliographic reference

4. The editing of the bibliography must be checked well to comply with the editorial rules.

5. Citation no. 45 (line 817) it is not clear what it is referred to.

6. Some parts left in Romanian need to be deleted or translated (lines 606-607).

7. The conclusions could be improved and consider the results obtained in a more complex way.

8. Please correct in the table 2 the word Județul with the word County

9. Please correct in the table 5: km2

In the paper you do not analyze the potential positive effect of the presence of the three National Parks in the region:

(Parcul naţional Cheile Nerei - Beușnița)

(Parcul naţional Semenic - Cheile Carașului)

(Parcul naţional Domogled - Valea Cernei)

WHY?

10. Please correct the line 269 “tourist intensity (Bulgaria and Serbia) or or 3.5 times that of Slovenia and 6.4 times”

11. Please correct the line 794

Refences

Number 32. RNDR, Bune Practici, 2014, No. 4 Anul II, USR, Departamentul Publicaţii MADR, disponibil pe: http://madr.ro (accesat în 16 Februarie 2023).

Regards

Author Response

Dear Reviewer,2

First of all thank you for time, spend to review our paper, then for your kind observation and proper suggestions. Your proper suggestions were beneficial to help us order/direct our scientific paper work.   So we re-organized our paper, more logically as you recommended us, in the following we will mention the main changes, highlighted with red color in the paper:

  • We consider valuable your suggestion regarding the text between lines 54 and 70, and we implement your suggestion, lines 72-99;
  • We also implement your valuable suggestion concerning Paragraph III. Cultural tourism including in a single table the text between lines 445 and 511, lines 611, table 11;
  • As far as concerning Paragraph V. Ethno-cultural tourism is also too long, we reduce it by using a table, lines 648, table 12;
  • Regarding your comment about spacing and punctuation we fixed this aspect;
  • Indeed it is advisable to always put the value in euros, so we did that, lines 436,511,522;
  • Regarding paragraph 3.2 we did not use a bibliographic reference because it is our ideas, our description;
  • Regarding the citation no. 45 it is Iancu, T.; Petre, I.L.; Tudor, V.C.; Micu, M.M.; Ursu, A.; Teodorescu, F.-R.; Dumitru, E.A. A Difficult Pattern to Change in Romania, the Perspective of Socio-Economic Development. Sustainability2022, 14, 2350, it is cited now, lines 827;
  • We did try to improve the conclusions, as better as we could, lines 817-827, 844-864, ;
  • We correct in the table 2 the word Județul with the word County;
  • We correct in the table 5: km2
  • Yes indeed we don’t analyze the three parcs because we concentrate on rural forms of tourism, and we already have a lot of material, but we did justify at lines 573-586 and remember those three parcs at lines 670-676; We did make the correction “tourist intensity (Bulgaria and Serbia) or or5 times that of Slovenia and 6.4 times”, line 381;
  • We make the correction from line 794;
  • We rephrase the reference from the number 32.

So in the end we hope we've been able to implement all your requirements. Thank you for your valuable suggestions and the time dedicated to reviewing our paper. You help us to direct our study on a better and much improved line.

The authors

Reviewer 3 Report

This research discusses a meaningful topic - different forms of sustainable tourism in Romania. My suggestions are below:

1. The title of this manuscript is "development possibilities analysis", and it is also a keyword listed, but "development possibilities" cannot be found throughout the manuscript. The authors should explain the meaning of this analysis because it sounds more like a special analysis method. Or I suggest they revise it to "analyzing the development possibilities of...."

2. Some parts of the manuscript read more like bullet points, which should be avoided in academic writing. For example, on page 2 there are several principles "- Minimizing the impact of tourist activity." The authors can rephrase these parts. Explaining them instead of listing them.

3. For the case study, the authors should explain why it is a good case to be analyzed. It will give insights into the contribution.

4. As the data is from before 2019, which is 4 years ago. Will the pandemic change the results? Some recent developments should be discussed.

I hope the above suggestions are helpful. Good luck!

Author Response

Dear Reviewer,

First of all thank you for time, spend to review our paper, then for your kind observation and proper suggestions. Your proper suggestions were beneficial to help us order/direct our scientific paper work.   So we re-organized our paper, more logically as you recommended us, in the following we will mention the main changes, highlighted with red color in the paper:

  1. We took in consideration your suggestion about the title, and we change it according to your suggestion, line 1-2;
  2. Regarding your request to bullet point, we improve the specified paragraph from page 2 there are several principles "- Minimizing the impact of tourist activity.", at lines 71-98;
  3. As far as concerning your suggestion about explain why it is a good case to be analyzed, we did that at line 239-251;
  4. Indeed the data is from before 2019, which is 4 years ago. We collect the data fron National Institute of Statistics, and 2019 is the last year published by the institute. This the metrics of data appearance in our country.

         So in the end we hope we've been able to implement all your requirements. Thank you for your valuable suggestions and the time dedicated to reviewing our paper. You help us to direct our study on a better and much improved line.

The authors

Reviewer 4 Report

Upon reading the manuscript, I have observed that while it contains potentially useful information, it lacks proper methodology and theoretical contribution. As a result, it cannot contribute significantly to the existing literature and is not suitable for publication in the Sustainability Journal.

I suggest revising the manuscript to address these issues and resubmitting it for further consideration.

Author Response

Dear Reviewer,

First of all thank you for time, spend to review our paper, then for your kind observation and proper suggestions. Your proper suggestions were beneficial to help us order/direct our scientific paper work.   So we re-organized our paper, more logically as you recommended us, in the following we will mention the main changes, highlighted with red color in the paper:

  • Concerning your remark about background for the tourism topic we have make some adings to sustain the benefits of this field, lines 145-151,55-59, 72-99;
  • As far as concerning the comment about the research support agritourism at lines 240-252 we did mention the importance of this field of tourism for Caras Severin area;
  • Regarding the impact over the environment was discussed at lines 113-127;
  • Regarding the support of agritourism against climate change impacts and support biodiversity and more resilient rural landscape, we did implement this suggestion in conclusion part, at lines 815-825 and 842-862;
  • As far as concerning the effectiveness of the proposed method we discussed it and justify it at lines 160-184.
  • We consider valuable your suggestion regarding the text between lines 54 and 70, and we implement your suggestion, lines 72-99;
  • We also implement your valuable suggestion concerning Paragraph III. Cultural tourism including in a single table the text between lines 445 and 511, lines 611, table 11;\
  • As far as concerning Paragraph V. Ethno-cultural tourism is also too long, we reduce it by using a table, lines 648, table 12;
  • Regarding your comment about spacing and punctuation we fixed this aspect;
  • Indeed it is advisable to always put the value in euros, so we did that, lines 436,511,522;
  • Regarding paragraph 3.2 we did not use a bibliographic reference because it is our ideas, our description;
  • Regarding the citation no. 45 it is Iancu, T.; Petre, I.L.; Tudor, V.C.; Micu, M.M.; Ursu, A.; Teodorescu, F.-R.; Dumitru, E.A. A Difficult Pattern to Change in Romania, the Perspective of Socio-Economic Development. Sustainability2022, 14, 2350, it is cited now, lines 827;
  • We did try to improve the conclusions, as better as we could, lines 817-827, 844-864, ;
  • We correct in the table 2 the word Județul with the word County;
  • We correct in the table 5: km2
  • Yes indeed we don’t analyze the three parcs because we concentrate on rural forms of tourism, and we already have a lot of material, but we did justify at lines 573-586 and remember those three parcs at lines 670-676; We did make the correction “tourist intensity (Bulgaria and Serbia) or or5 times that of Slovenia and 6.4 times”, line 381;
  • We make the correction from line 794;
  • We rephrase the reference from the number 32.
  • We took in consideration your suggestion about the title, and we change it according to your suggestion, line 1-2;
  • Regarding your request to bullet point, we improve the specified paragraph from page 2 there are several principles "- Minimizing the impact of tourist activity.", at lines 71-98;
  • As far as concerning your suggestion about explain why it is a good case to be analyzed, we did that at line 239-251;
  • Indeed the data is from before 2019, which is 4 years ago. We collect the data fron National Institute of Statistics, and 2019 is the last year published by the institute. This the metrics of data appearance in our country.

So in the end we hope we've been able to convince you that it was a lot of work of doing this research, we did make many changes so as to be suitable for publication in the Sustainability Journal. Thank you for the time dedicated to reviewing our paper.

The authors

Reviewer 5 Report

I would like to thank the authors for presenting a very interesting topic of research in Romania. The title of the paper is fully adequate. The content of the abstract is noticeable and very good and adequate keywords. The introductory part contains elements of an introduction and review of alliteration and I propose to strengthen the literature review. The results are displayed very clearly. It is about statistical data presented through figures. 1 The figure could have a higher resolution, to be a cleaner picture. I suggest strengthening the text about the future implications of the research and results, as well as references:

Gajić, T.; Blešić, I.; Petrović, M.D.; Radovanovć, M.M.; Đoković, F.; Demirović Bajrami, D.; Kovaˇci´c, S.; Jošanov Vrgovi´c, I.; Tretyakova, T.N.; Syromiatnikova, J.A. (2023). Stereotypes and Prejudices as (Non) Attractors for Willingness to Revisit Tourist-Spatial Hotspots in Serbia. Sustainability,15, 5130. https://doi.org/10.3390/su15065130

Gajić, T.,Vukolić, D., Petrović, M., Blešić, I., Zrnić, M., Cvijanović, D., Sekulić, D., Spasojević, A., Obradovi, A., Obradović, M. , Savić, I., Jovanović, J., Gajić, M., Lukić, D., & Anđelković, Ž. (2022). Risks in the Role of Co-Creating the Future of Tourism in "Stigmatized" Destinations. Sustainability 2022, 14, 1-19, 15530. https://doi.org/10.3390/su142315530

After corrections it is possible to publish

Author Response

Dear Reviewer,

First of all thank you for time, spend to review our paper, then for your kind observation and proper suggestions. Your proper suggestions were beneficial to help us order/direct our scientific paper work.   So we re-organized our paper, more logically as you recommended us, in the following we will mention the main changes, highlighted with red color in the paper:

  • We appreciate favorable your suggestion about the future implication and consequences and we did analyze the impact of tourist activity on the natural environment at line 71-98;
  • As far as concerning the references specified we did used them.

So in the end we hope we've been able to implement all your requirements. Thank you for your valuable suggestions and the time dedicated to reviewing our paper. You help us to direct our study on a better and much improved line.

The authors

Round 2

Reviewer 2 Report

Dear Authors,

thank you for your kind answer.

I think you need to check the punctuation of the parts added in red.

For the rest the work has improved and has reached a good level of reading.

Thanks for your efforts.

Best regards

Author Response

Dear Reviewer
First of all thank you for time, spend to review our paper, then for your kind observation and proper
suggestions. Your proper suggestions were beneficial to help us order/direct our scientific paper work.
✓ We kindly thank you for your concern, we did check and fixed the punctuation of the parts
added in red
Thank you for your valuable suggestions and the time dedicated to reviewing our paper. You help us
to direct our study on a better and much improved line.
The authors

Reviewer 4 Report

The revised version of the article has shown some improvement. However, there are still some minor corrections that need to be made before it can be accepted for publication. The conclusion has been improved, but it would benefit from more in-depth discussions. Furthermore, the practical and theoretical implications of the research should be presented more clearly to enhance the article's scientific rigor.

Author Response

Dear Reviewer,
First of all thank you for time, spend to review our paper, then for your kind observation and proper
suggestions. Your proper suggestions were beneficial to help us order/direct our scientific paper work.
So we re-organized our paper, more logically as you recommended us, in the following we will mention
the main changes, highlighted with red color in the paper:
✓ Concerning your remark about the conclusion we presented the practical and theoretical
implications of the research, lines 890-900.
So in the end we hope we've been able to convince you that it was a lot of work of doing this research,
we did make many changes so as to be suitable for publication in the Sustainability Journal. Thank you
for the time dedicated to reviewing our paper.
The authors
